# Analysis of Policies to Protect the Health of Urban Refugees and Asylum Seekers in Thailand: A Qualitative Study and Delphi Survey

**DOI:** 10.3390/ijerph182010566

**Published:** 2021-10-09

**Authors:** Sataporn Julchoo, Mathudara Phaiyarom, Pigunkaew Sinam, Watinee Kunpeuk, Nareerut Pudpong, Rapeepong Suphanchaimat

**Affiliations:** 1International Health Policy Program, Ministry of Public Health, Nonthaburi 11000, Thailand; mathudara@ihpp.thaigov.net (M.P.); pigunkaew@ihpp.thaigov.net (P.S.); watinee@ihpp.thaigov.net (W.K.); nareerut@ihpp.thaigov.net (N.P.); rapeepong@ihpp.thaigov.net (R.S.); 2Sirindron College of Public Health, Chonburi 20000, Thailand; 3Department of Disease Control, Division of Epidemiology, Ministry of Public Health, Nonthaburi 11000, Thailand

**Keywords:** refugees, urban refugees, asylum seekers, health protection, health promotion, well-being

## Abstract

The health of urban refugees and asylum seekers (URAS) in Bangkok has been neglected and health policies for USAR have not materialized. This study aimed to explore the views of stakeholders on policies to protect URAS well-being in Thailand. This study conducted a mixed-methods approach comprising both in-depth interviews and Delphi survey. The interview findings revealed six main themes: (1) the government position on URAS; (2) opinions on Thailand becoming a party of the 1951 Refugee Convention; (3) NGOs on health promotion for URAS; (4) options on health insurance management for URAS; (5) working potential of URAS; and (6) uncertainty of future life plans for URAS. The Delphi survey showed that URAS should have the right to acquire a work permit and be enrolled in the public insurance scheme managed by the Ministry of Public Health. Moreover, the ideology of national security was more influential than the concept of human rights. The ambiguity of the central authorities’ policy direction to take care of URAS creates haphazard legal interpretations. The Delphi survey findings suggested the need for a more inclusive policy for URAS, however actual policy implementation requires further research on policy feasibility and acceptance by the wider public.

## 1. Introduction

As of 2019, the estimated number of migrants internationally amounted to 272 million, equivalent to about 3.5% of the world’s population [1]. In 2019, there were 79.5 million forcedly displaced people around the world. Among these, 26 million were refugees, and 3.5 million were asylum seekers [2]. The volume of refugees worldwide has greatly increased from 10 million to 26 million in the last decade. Approximately 68% of refugees globally originated from five countries, including Afghanistan, Myanmar, South Sudan, Syria, and Venezuela. The major refugee crises that contributed to massive displacement of people were the exodus of the Rohingya people from Myanmar to Bangladesh and the conflict in Arab countries that caused the unprecedented outflow of refugees from countries including Syria, Iraq, and Libya into Europe [2].

Refugees and asylum seekers usually face many health threats, including infectious diseases, non-communicable diseases, and mental health [3]. For infectious diseases, refugee and asylum seekers are likely to be more vulnerable to serious outbreak because of poor living conditions, poor sanitization, and lack of access to healthcare [4,5,6,7]. Kondilis et al. demonstrated that, during the coronavirus disease 2019 (COVID-19) pandemic, refugees and asylum seekers in Greece faced numerous events of outbreaks. The overall 9-month incidence of COVID-19 amongst refugees and asylum seekers in Greece was relatively high (about 2000 cases per 100,000 population) [8].

Thailand is among the most common destination for cross-border migration in Southeast Asia. The majority of migrants in Thailand came from its neighboring countries, namely, Cambodia, Lao PDR, Myanmar, and Vietnam (so-called CLMV nations). At present, there are more than 3 million migrants living in Thailand [9].

In addition, Thailand is also a residence for refugees and asylum seekers [10]. The country has hosted refugees along the Thai–Myanmar border for more than four decades. During the 1980s and 1990s, Thailand faced a huge influx of refugees from Myanmar because of the conflict between ethnic groups and the Myanmar government. To date, there are nine temporary shelters for almost 100,000 refugees [10]. Apart from sheltering refugees along the border, Thailand is a host country for urban refugees and asylum seekers (URAS). The majority of URAS live in the capital city, Bangkok, approximately 5000. The well-being of sheltered refugees shows tangible advancement as it is relatively straightforward to implement a policy in a well-defined geographical space. The United Nations High Commissioner for Refugees (UNHCR) and international non-governmental organizations (NGOs), such as Médecins Sans Frontières and the International Rescue Committee, provide additional humanitarian assistance in the shelters. In contrast, the health of URAS in Thailand has not been widely discussed in most policy dialogues [11]. In addition, almost all URAS live scattered across the city, creating difficulty in identifying a main responsible agency to take a pivotal role to protect the health of URAS at the utmost. A greater understanding about necessary health policies that support the health of URAS has important public health implications as URAS are necessarily involved with Thailand’s quest to achieve universal health coverage (UHC). The Thai government has set a clear direction for UHC where ‘all people’ on Thai soil must have their health protected. This is stipulated in many policy documents such as the Border Health Plan of the Ministry of Public Health (MOPH) (2017-2521) [12] and Strategies for National Health Insurance Development of the National Health Security Office (2017-2521) [13].

Therefore, the main objective of this study was to explore views of stakeholders engaged with policies that are related to or have influenced the health and well-being of URAS in Thailand. We also investigated the views of URAS themselves to complement the stakeholders’ perspectives.

## 2. Methods

### 2.1. Study Design and Setting

This study employed a mixed-methods approach, comprising both qualitative and quantitative data collection, and focusing on URAS in Greater Bangkok only (*N*~5000).

### 2.2. Data Collection Methods

For the qualitative strand, we used in-depth interviews. Prior to and right after the fieldwork, we arranged a meeting among the team members to finetune the understanding on the interview topics and data from the interviewees. Each interview took approximately 45–60 minutes per informant. For Thai interviewees, the interviews either took place face-to-face in the workplace of the informant, or, during a telephone interview, in a private room. For URAS informants, we used face to face interview at the office of Bangkok Refugee Center (BRC). BRC is the main NGO under the patronage of UNHCR. The main function of the BRC is to provide legal support and counselling for URAS. The BRC staff also assisted the research team to recruit URAS who could serve as the informant for the study. However, the BRC staff did not present themselves during the interview with URAS. Only the researchers and URAS informants presented during the interview in BRC private room. A purposive sampling was used to identify key informants and additional informants were identified by snowball selection. The sampling of Thai informants focused on those who had been involved in the policies for URAS or had ever dealt with the research on URAS.

The interviews were audio-recorded and transcribed verbatim, with consent from the interviewees. Key information from the interviews was used to guide the Delphi survey’s questions. The total number of 37 interviewees consisted of nine representatives from NGOs with work experience on URAS, five representatives from the MOPH, five independent academics, four healthcare providers in public health facilities where URAS usually visited, four policymakers from the National Health Security Office (NHSO), Ministry of Labor (MOL), Ministry of Foreign Affairs (MFA), and Ministry of Interior (MOI), three representatives from international development agencies, and seven URAS from six nationalities (Afghani, Iraqi, Pakistani, Somali, Sri Lankan, and Vietnamese), Table 1.

For the quantitative strand, a Delphi survey was exercised. We invited the informants from the prior in-depth interviews to take part in the Delphi survey. However, only thirteen informants agreed to participate (four NGO staff, three policymakers, two representatives from international development agencies, one academic, and one healthcare provider). The survey began by circulating the questionnaire via an electronic mail to these thirteen participants. We then collected their responses and sent the questionnaire back to them for another two rounds. In the later rounds, the participants were able to view the group’s responses. Two participants dropped out in the later rounds. This meant, finally, eleven participants completed the three-round survey. Once the survey was completed, we then re-interviewed or had an informal discussion with some interviewees to triangulate the survey results against the survey responses (and vice versa) and to assess if any additional themes would emerge. The data were collected in the office computer with password protection. Only the principal investigator could access the interview data. The linkage between the interview and the Delphi survey is visualized in Figure 1.

### 2.3. Interview and Survey Topics

The interview and Delphi survey topics were based on the following framework, Figure 2.

We adapted the concept of the social determinants of health [14] to construct the above framework. In the field practice, the interviews started by building rapport with the informants. For the interviews with the non-URAS informants, researchers emphasized the following issues: overarching policy direction of the government towards the care for URAS (such as international relationships and politics, and international laws and regulations ratified by the Thai government); existing legal mechanisms and policy instruments for URAS (such as immigration law, nationality law, and the present health insurance scheme); and the views and attitudes of the informants towards the optimal approach the Thai government could take towards URAS’s health. When we interviewed URAS, we focused mostly on their experiences in accessing health care in Thailand and factors that influenced their well-being.

The Delphi survey questionnaire consisted of twenty statements in four domains, namely: (i) health financing; (ii) benefit package; (iii) health insurance; (iv) policies to support aspects of well-being (such as education and work rights); and (v) policy direction of the Thai government. A list of all twenty statements is presented in Appendix A. The participants were asked to rate from one (least agree) to five (most agree), noting if and to what extent they agreed with each statement.

### 2.4. Data Analysis

For in-depth interviews, inductive thematic content analysis was exercised. The researchers began by familiarizing themselves with the interview data, reading through the transcriptions and fieldwork memos and listening to the interviewed audios. Keywords and sentences were highlighted and those with similar content were labeled with the same color. Then, several codes with relevant contents were grouped together and merged into theme. We were able to identify six main themes from the interviews. The coding was completed by SJ and RS. If there were any contradictory issues between the two coders, a consultation with a public health expert of the filed would be performed. The manual coding was performed. Microsoft Excel was used to store the quotes. We presented a coding tree in Appendix A. The interview data were triangulated with the field note and policy documents if needed. Before the final report was published, the researchers arranged a stakeholder meeting on 17 August 2020 to ask for feedback or comment on the findings. For the Delphi survey, descriptive statistics were used. The informants’ response was shown in terms of median and percentile.

### 2.5. Ethical Consideration

This study obtained ethics approval from the Institute for Human Research Protection, Thailand (IHRP 595/2562). The data collection process of this study strictly followed the Declaration of Helsinki. The informed consent process in this study was firmly approved by IHRP. All informants were given the participant information sheet before the interview and the survey. Written consent was obtained from all Thai informants. For URAS informants, we accepted verbal consent instead of written consent to avoid any sense of coercion. We offered the Thai informants USD 32 each to compensate for their time dedicated to the interview. For URAS, we provided each interviewee a thank you gift of about USD 10 after the interview was completed.

## 3. Results

### 3.1. Themes Identified from the Interviews

We identified six main themes from the interviews: (i) the Thai Government position on URAS; (ii) opinions on Thailand becoming a party of the 1951 Refugee Convention; (iii) non-government organizations on health promotion for URAS; (iv) options on health insurance management for URAS; (v) working potential of URAS; and (vi) uncertainty of future life plans for URAS.

#### 3.1.1. Thai Government Position on URAS

All stakeholders suggested that the Thai government’s position in taking care of URAS was related not only to the healthcare and well-being policies, but also encompassed issues of wider national security and international relations. One of the interviewees pointed out that the Thai government had a clear position to be ‘unclear’. If the government’s position was too open and supported a human rights concept, there would be concerns that the country could attract more URAS to enter Thailand.

‘This is a tricky policy of Thailand…. Let it all happen. This is a very interesting point when you talked about international relations. I thought the Thai government might know that the government will be in trouble if they are too stiff. So just ignore it, then there will be an excuse.’(C2)

Some key informants (A1, B2, and C4) identified that the ambiguity of the overarching policy direction towards URAS caused incoherent practices towards the care for URAS among frontline officers, and created a situation where there was no clear agency accountable for the care for URAS.

‘In the other sectors, I don’t know if they have main officers to take care of urban refugees and asylum seekers. But in the health sector, we don’t have any main responsible agencies.’(C4)

Health sector stakeholders (B2, B3, and B5) commented that health services, such as health insurance for URAS, should be free from the limitations caused by national security concerns. However, interviewees who had work experience with the national security sector (C3 and E2) argued that an insurance policy for URAS should be launched but performed covertly.

‘This topic is politically sensitive. If the MOPH thinks that health insurance is necessary for them (URAS), we can implement it but this must be done unofficially.’(B3)

#### 3.1.2. Opinions on Becoming a Party of the 1951 Refugee Convention

There were diverse comments about Thailand becoming a party of the 1951 Refugee Convention. Some stakeholders (C3 and E1) commented that the 1951 Refugee Convention committed Thailand to undertake more works for URAS, yet other agencies that supported URAS, such as UNHCR, were not committed to provide support for Thailand. Moreover, some stakeholders noted that UNHCR was not performing well in taking care of URAS. The UNHCR migrant-screening program was not effective enough to grant refugee status for those really in need.

‘In the national security view, becoming a signatory of the Convention (1951 Refugee Convention) is not the best choice.’(E1)

Some interviewees (C1 and F2) commented that Thailand would obtain benefits if the country became a party of the convention. Those gaining substantial benefits from becoming a party were NGOs and academics who could use the convention as a tool to drive the healthcare policy agenda for URAS. For some, the 1951 Refugee Convention did not differ from other conventions that Thailand has signed which also promote human rights.

‘In my opinion, the 1951 Refugee Convention is about the protection of refugee rights. This topic appears in international laws. The Thai government has already signed many other conventions that are related to human-rights protection. Therefore, I do not see any difference (if Thailand signs the 1951 Refugee Convention).’(F2)

One of the stakeholders (F3) mentioned that although the Thai government was not part of the 1951 Refugee Convention, the country performed quite well in the care for URAS. The signing of the convention would be an optional benefit, though not necessary.

‘I think Thailand has always been a silent country, even though Thailand has not signed the Refugee Convention. But still the fact remains that there have been refugees in Thailand for many years. And I would say we have been handling the refugee situation quite positively.’(F3)

#### 3.1.3. Non-Government Organizations on the Health Promotion for URAS

This theme is linked to the unclear policy direction of the Thai government towards URAS because when officials did not have a clear mandate to deal with URAS, a few NGOs and civic groups stepped in. BRC is amongst a few charitable organizations responsible for the care for URAS. It provided health consultancy services for URAS. Some other functions included home visits, provision of cash-based interventions (approximately USD 96–125 per month) and subsidizing basic medical expenses, especially for children and pregnant women.

‘We support vaccination for children under five years and children with health complications and we also cover the medical expense…We may ask for support from the hospital if the health condition is severe and creates huge medical expenses.’(A3)

Other NGOs, in addition to the BRC, also intervened. The Buddhist Tzu Chi Foundation organized a clinic for URAS once a month and Asylum Access Thailand (AAT), Center for Asylum Protection (CAP), and Jesuit Refugee Service (JRS) provided legal advice for asylum seekers seeking refugee status. Good Shepherd Bangkok arranged a language school for URAS children, although the school was not authorized by the Ministry of Education. These organizations worked in cooperation with each other, although there was no formal agreement among them.

#### 3.1.4. Options on Health Insurance Management for URAS

The MOPH has implemented a public insurance scheme, namely, the “Health Insurance Card Scheme” (HICS) for Cambodia, Lao PDR, and Myanmar (CLM) migrant workers for more than a decade. The scheme provides comprehensive medical benefits at registered public hospitals. The applicant needs to pay an annual premium, including the health check expense, for about USD 64 [15]. In contrast to the HICS for CLM migrant workers, there has been no public health insurance for URAS so far. In the past there was an attempt to widen the legal interpretation of the HICS to allow URAS to buy the insurance but this failed.

Some interviewees (A2, A3, A4, C3, and E3) mentioned that the Cabinet Resolution on 15 January 2013, by document, did not specify eligible nationalities to the insurance buyer. Therefore, at that time, some hospitals allowed URAS to buy the insurance but then sales were later cancelled due to unclear direction from the Thai government. Some hospitals found that those who bought the insurance were mostly children and elderly people who were prone to sickness.

‘We found that most urban refugees were uninsured. Some NGOs said that some URAS were able to purchase the health insurance in the past but this was cancelled because of unclear communication.’(A6)

Some stakeholders (for instance, B1 and G1) mentioned that if URAS were allowed to work, they could have the right to be enrolled in the insurance.

One of the participants (F3) highlighted that health insurance for URAS should not be separated from existing insurance schemes, according to the concept of an ‘inclusive policy’ and effective pooling of risk.

‘First of all, when we talk about health insurance, it will not work when we divide it into different categories, right? The bigger the (beneficiary) pool is, the better the survival of the insurance scheme is, right?’(F3)

However, healthcare providers (D2 and D3) commented that health insurance for URAS should not be part of the UHC policy as it might consume healthcare resources that belonged to Thai citizens. Additionally, if any medical expenses incurred, these should be covered by UNHCR.

‘UNHCR is larger than our hospital. Why should the financial support for refugees be our responsibility? UNHCR should support all of them. If your guest overstays in your home and they do not pay electronic bills, food expenses, and medical expenses, how do you think about this? Will you pay for them?’(D3)

#### 3.1.5. Working Potential of URAS

All of the themes above viewed URAS as service users. However, a few interviewees (B1, D4, and E3) suggested that URAS had work potential because some URAS used to work as professionals. Granting URAS the right to work would benefit the country’s economy and at the same time decrease public expense.

‘If URAS reside in Thailand and do not cause any social problems, we must have information about them, about their residence and we should allow them to work and purchase health insurance.’(E3)

Some URAS interviewees informed us that they could speak many languages and used to work as translators in a range of organizations. Some URAS had tried to find a job online and some had graduated with a degree in their previous country of residence. However, the employment policy for non-Thais does not allow URAS to acquire a work permit.

‘(ASK: What’s your job now?) I work at home. I find a movie and then translate I the movie. I translate it from English to my own language. I like keeping myself busy because when I’m free, my thoughts get worse. There is lots of negative thinking in my mind. Then I do not feel good.’(G3)

#### 3.1.6. Uncertainty of Future Life Plans of URAS

All URAS interviewees informed that they did not have any clear long-life plans. They hoped for a resettlement in a third country, but if such a plan was not feasible, they wished to continue living in Thailand.

‘I was hoping, you know, to get freedom, to start a really new life as Thai people do…Or that we can work to start our life. I have plan to start education if I can, but I know I cannot.’(G2)

Some of the interviewees (E2 and F2) commented that the process of resettlement was extremely long and arduous. At the same time, the process for settlement in Thailand was problematic. Some URAS entered the country as a tourist but lived here even after the visa expired.

### 3.2. Delphi Analysis Results

The Delphi survey among 13 experts found that the issues that the participants agreed upon most were that URAS should have the right to work, the right to buy MOPH health insurance or insurance provided by the private sector, the right to access basic education, the right to live outside the detention centers, and the right to receive medical benefits that cover treatment for public health threats and conditions (e.g., tuberculosis and influenza). All of these issues received the median score of five. Issues that most experts rated the least (median = two) were; ‘Overall, the Thai government presents appropriate policy direction towards URAS’, ‘The medical benefit for URAS should focus on emergency illnesses and accidents only’, and ‘The medical charge of URAS should be mainly shouldered by NGOs’, Figure 3, Figure 4, Figure 5, Figure 6 and Figure 7.

## 4. Discussion

The study is among the very first studies in Thailand, which focus on the country policy and the attitudes of all relevant stakeholders towards the well-being of URAS.

The interviews and Delphi-survey suggested that the ideology of national security and international relations influenced the concept of human rights for URAS. The concept of URAS health has been tightly woven with foreign policy issues over many years. Health issues are still considered “low politics”, while foreign policy is deemed to be “high politics” in decision making [16,17]. This situation has occurred not only in Thailand, but also in many other parts of the world. For example, according to a study by Klaus, the campaign for the restriction of rights for immigrants and refugees in Poland was used by the populist political parties to win the election in 2015 [18]. By late 2015, the Polish government launched the Antiterrorist Act stipulating that every foreign citizen would be put under surveillance without any court control with an ultimate aim to stop a refugee influx into the Polish territory.

With regard to URAS problem solving mechanism, the UN agencies usually puts emphasis on the “relief” of URAS’ suffering rather than addressing the structural problems that compromise the well-being of migrants. The same situation often occurs with NGOs or charitable organizations that most of the time serve as humanitarian support for URAS. This problem reflects the limitation of UN agencies that do not have solid legal mechanisms to force or even encourage any particular nation to re-orientate its health system by making it more “inclusive” for all people on its soil. Therefore, a more practical approach for UN agencies and NGOs is to comply with the local government. However, this approach has a setback as most of the NGOs’ (or even UNHCR’s) activities usually end up with a humanitarian relief for URAS (or any activities that make URAS more “resilient” with the status quo health system) as it is less disputable compared with shaking the structural problems or recognizing URAS as part of the society on equitable grounds with the nationals [19].

Although Thailand is not a signatory to the 1951 Refugee Convention, the country has long been a key member of ASEAN [20], which has its own agreement among member states to guarantee human rights, including the right to health for everybody in the region. Therefore, Thailand cannot deny its responsibility to protect the health of URAS. However, it appears that current Thai laws are not developed to protect URAS from the outset. The majority of byelaws for non-Thais mostly concern migrant workers, but not URAS. The most relevant law relating to URAS is the ‘Regulation of the Office of the Prime Minister: Screening process for aliens entering the Kingdom of Thailand and incapable of returning to their home country’ B.E. 2562 (2019). It establishes a screening mechanism for groups of aliens in line with the nature of Thai society and international situations in order to reach sustainable solutions for Thailand’s alien management problem [21].Yet, the regulation still lacks operational details. Whether Thailand will become a party of the 1951 Refugee Convention or not may not be so important, what may be more important is to have a clear stance on the Thai government’s responsibility to take care of URAS.

As long as there is no clear direction from central authorities, there will always be variation in day-to-day operations of the treatment of URAS among street-level bureaucrats [22]. A clear example can be noticed in the recent COVID-19 pandemic in Thailand. Though the Thai government publicly announced to the wider public that all COVID-19 patients (regardless of the nationalities) are able to access free treatment of COVID-19. The hospitals are able to reimburse the healthcare cost from the MOPH and the National Health Security Office (NHSO). However, in practice, there are also administrative problems in reimbursing the healthcare cost for undocumented migrants, not to mention URAS [23,24].

If the Thai government steadfastly rejects the accession of the 1951 Refugee Convention, the domestic mechanisms to take care of URAS health should be strengthened. Moreover, the Thai government should learn from international experience, especially from countries that accepted a refugee influx while refraining from the 1951 Refugee Convention. Janmyr raised the case of Lebanon as an example of a country that is not a party of the 1951 Refugee Convention, but is hailed by the international community for its generosity towards refugees as probably hosting the highest number of refugees in the world in proportion to its population size [25]. Lebanon already has human rights obligations towards refugees on its territory by virtue of membership of the United Nations and its ratification of a number of core human rights instruments [25]. Another interesting issue is the URAS right to work and eligibility to be enrolled in public health insurance in Thailand. For the right to work, Brown et al. suggested that the lack of legal instruments to allow URAS to work legitimately has caused URAS to face a greater risk of being arrested or detained [26].

Additionally, the provision of the right to work is likely to lead to better quality of life for URAS because they will be able to access the labor market and gain sustainable livelihood opportunities. The right to work will also allow URAS to gain increased self-reliance and dignity and improve mental health [27,28]. Fleay and Hartley cited a case study in Australia, suggesting that without the right to work, asylum seekers in Australian communities faced exacerbated feelings of anxiety, sadness, and fear [29]. Most of this study’s interviewees, including those who participating in the Delphi survey, indicated that the right to work should be implemented as soon as possible with no critical disagreement from the wider public. The law to promulgate this policy—Royal Ordinance Concerning the Management of Employment of Foreign Workers, B.E.2560 (2017)—is in place [30] and only a Cabinet Resolution is required to implement the policy concretely.

Giving URAS the right to work also means that Thailand will benefit from an increased labor force (and some URAS are quite well educated). The opening of employment opportunities helps improve the registration data of URAS and positively affects the Thai economy. Additionally, to deal with the refugee crises, it has been recommended that, while poorer countries accede to host refugees, richer countries should help provide financial support to those host countries in order to protect refugees’ health and well-being [31]. Since Thailand host numbers of refugees from various nations, this may benefit Thailand as well.

High healthcare costs and financial difficulties are also a key barrier faced by refugees and asylum-seekers [32]. According to Elsouhang et al. possessing health insurance was significantly associated with increased utilization of medical services among Iraqi refugees in the United States of America [33]. In the Thai context, the financial difficulties that URAS face are not just a matter concerning the work performance of an individual, but these also intertwine with the legal design of a system that does not allow URAS to work legitimately. Moreover, the inclusion of URAS in public health insurance schemes may benefit not only URAS, but the health system as a whole. A substantial volume of literature shows that the exclusion of non-national populations (such as migrants, refugees, and asylum seekers) from official primary health care might save costs early on, but this effect might be lost as costs are shifted to healthcare providers in secondary care or community settings [34]. Again, the promotion of the right to public health insurance for URAS is interlinked with the affordability of insurance, which is also linked to the right to work. It is also linked to the regulation of the state which needs to be clearly specified in laws or official state regulations in order to avoid haphazard interpretation by local healthcare providers [35].

Concerning methodological approaches, some limitations remain in this study. Firstly, this study did not encompass all types of refugees and asylum seekers. Those who were detained in detention centers and those living in the temporary shelter areas along the country border were excluded. A discussion on policies to take care of the health of these populations necessitates a more extensive review on the relevant laws and perhaps requires a wider range of interviewees (including police and state prosecutors). Secondly, the issue of healthcare for URAS is relatively sensitive. According to research ethics, in the fieldwork, the researchers needed to disclose their own work status as persons working with the MOPH, which may have created an unfavorable feeling among the URAS interviewees. For example, some URAS might feel uncomfortable disclosing their life stories or unpleasant experiences with state officials to the researchers. In contrast, state official interviewees may have tended to answer the interview questions in a way that tried to meet the researchers’ expectation (social desirability bias). However, the researchers addressed these issues by using methodological triangulations (such as contrasting the interview findings with the review on policy documents). Lastly, suggestions from the interviews or Delphi survey (for instance, the provision of rights to employment and public health insurance enrollment) does not mean final consensus from policy experts and of course does not mean that these suggestions can be implemented without societal disagreement. In reality, to implement such policies, there needs a much wider consultative process from all angles of the political sphere. Further studies that explore policy feasibility for URAS are of great value.

## 5. Conclusions

Policies to protect the health of URAS are mainly influenced by the ideology of national security and international relations and the concept of human rights is considered “low politics” in the power operations in all political spheres. The ambiguity of the policies relating to URAS coming from central authorities has caused varying legal interpretations and incoherence of practice among frontline officers regarding healthcare for URAS. The right to legitimate employment and public insurance enrollment are issues that should be implemented soon in order to improve URAS’s livelihoods. Further studies that examine the feasibility of implementing these proposals and the extension of studies to cover refugees and asylum seekers from non-urban settings, such as detainment centers or sheltered areas, are recommended.

## Figures and Tables

**Figure 1 ijerph-18-10566-f001:**
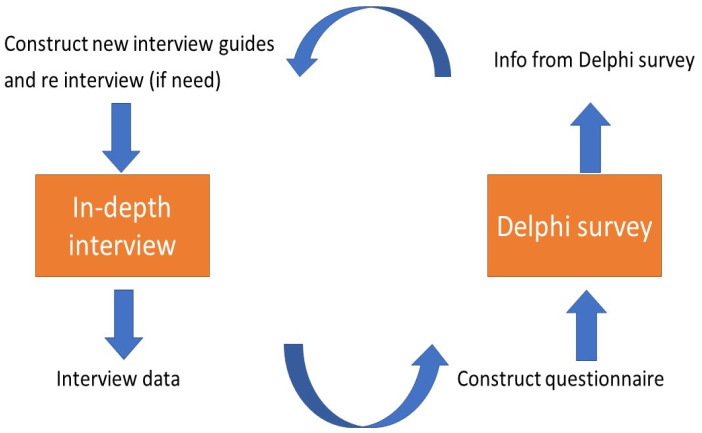
Linkage between in-depth interviews and Delphi survey.

**Figure 2 ijerph-18-10566-f002:**
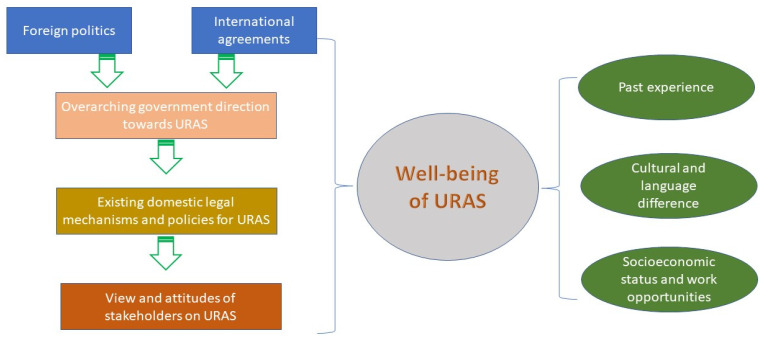
Conceptual framework serving as a basis for the interviews and the Delphi survey.

**Figure 3 ijerph-18-10566-f003:**
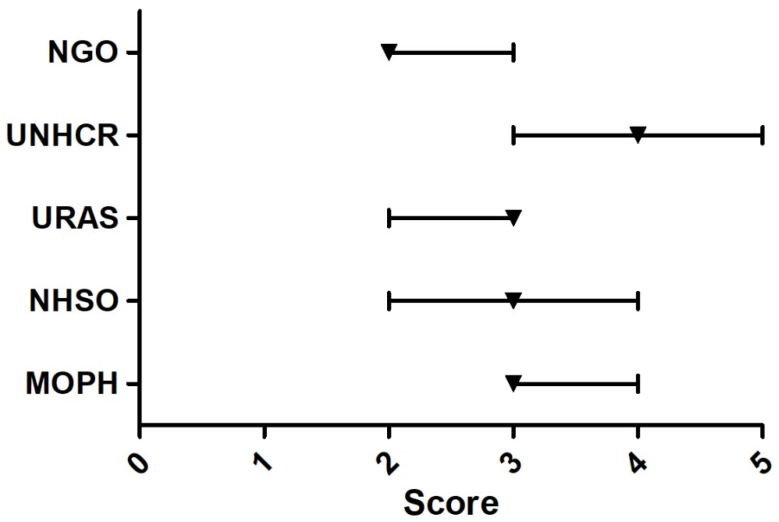
Delphi survey asking about the main responsible authority that should cover medical charges of URAS. Note: The square on the line denotes the median score. The left and right ends of the line denote the score at 25th and 75th percentile, respectively. MOPH = 3, NHSO = 3, URAS = 3, UNHCR = 4, NGO = 2.

**Figure 4 ijerph-18-10566-f004:**
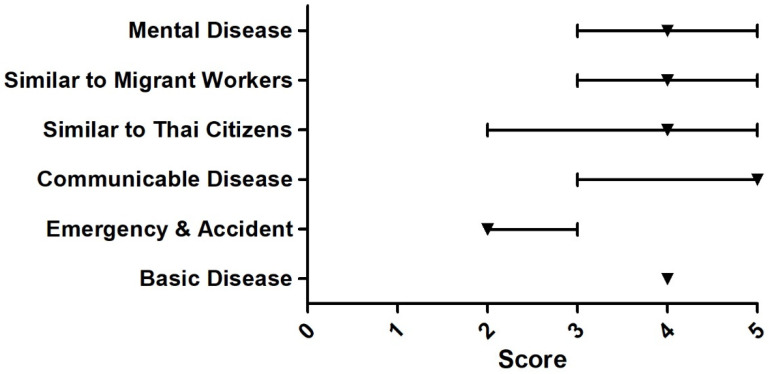
Delphi survey asking about the benefit package that should cover for URAS. Note: The Scheme 25th and 75th percentile, respectively. Basic disease = 4, emergency and accident = 2, communicable disease = 5, similar to Thai citizens = 4, similar to migrant workers = 4, mental disease = 4.

**Figure 5 ijerph-18-10566-f005:**
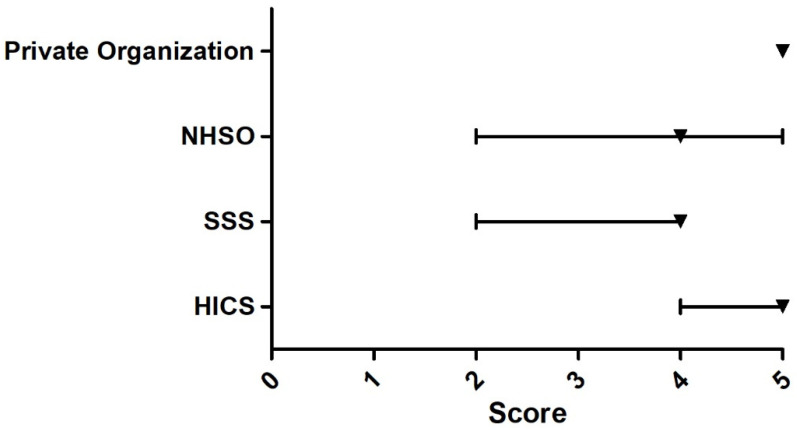
Delphi survey asking about the main health insurance provider that should cover for URAS. Note: The square on the line denotes the median score. The left and right ends of the line denote the score at 25th and 75th percentile, respectively. HICS = 5, SSS = 4, NHSO = 4, Private organization = 5.

**Figure 6 ijerph-18-10566-f006:**
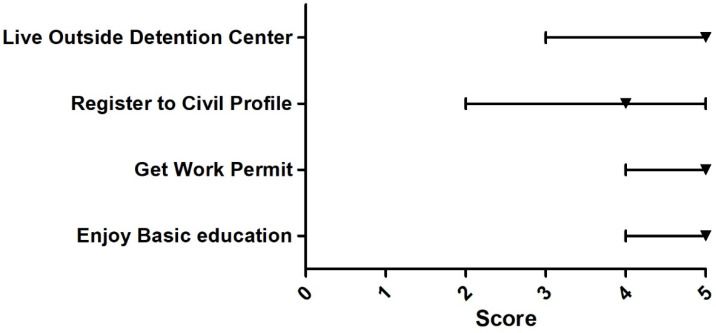
Delphi survey asking about the other aspects of well-being that should urban refugees should have rights. Note: The square on the line denotes the median score. The left and right ends of the line denote the score at 25th and 75th percentile, respectively. Enjoy basic education = 5, get work permit = 5, register to civil profile = 4, live outside detention center = 5.

**Figure 7 ijerph-18-10566-f007:**
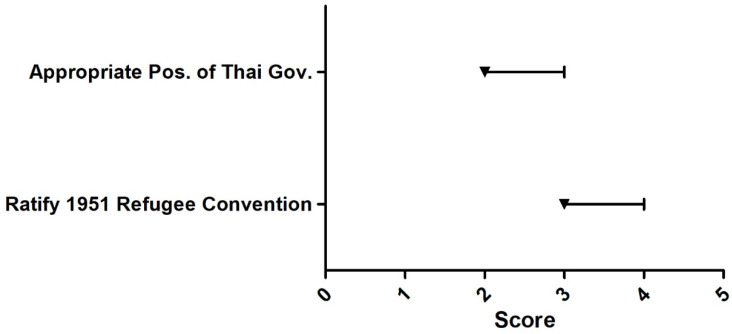
Delphi survey asking about position of Thai government towards URAS. Note: The square on the line denotes the median score. The left and right ends of the line denote the score at 25th and 75th percentile, respectively. Ratify to the 1951 Refugee Convention = 3, appropriate position of Thai government = 2.

**Table 1 ijerph-18-10566-t001:** Characteristics of key informants’ interviews.

Demographic	Key Informants (*N*)
Sex Male Female	19 18
Role and responsibility MOPH policy maker NSC, MOI, MOL, MFA policy maker International organization representers NGOs staff Academic Healthcare providers in public hospitals URAS from six nationalities	5 4 3 9 5 4 7

## Data Availability

Data available on request due to ethical restrictions.

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
