# Peer review of "Analysis of Policies to Protect the Health of Urban Refugees and Asylum Seekers in Thailand: A Qualitative Study and Delphi Survey"

_ijerph, 2021, doi:10.3390/ijerph182010566_

Round 1
Reviewer 1 Report
Referee report on "Health protection and well-being promotion for urban refugees and asylum seekers in Thailand: a quantitative study and Delphi survey"
The authors of the paper in front of us deal with an interesting question that immediately attracts attention of the reader. However, their analysis is very far from adequate. The authors run two surveys. For one of them, they ask 37 different people, while for the other, deeper, questionnaire they are left with only 11 people. The answers they receive are intuitive, but my major concern is what an external reader could learn from reading these answers? Moreover, the authors also do not explain how they select some of their interviewees. For example, they mention 5 academics, whom they interview, but what is the reason why these persons were chosen? Similarly, as I understand, they are interested in URAS from Myanmar, but they interview 6 persons from 5 other countries, while no person from Myanmar was chosen to participate. The authors report some of the answers they received, but I do not see any academic analysis of these answers.
I can attract the authors to many detailed points, but let me mention just one of them: Even in the very first sentence of their text, they make a computational mistake, when they write that “as of 2019, the estimated number of migrants internationally amounted to 272 million, equivalent to about ONE percent of the world population”. I would recommend the authors to make the computations once again.
Overall, the style of this piece is more appropriate to a popular magazine, rather than to an academic journal.
Author Response
Response to Reviewer 1 Comments
Please see the attachment

Reviewer 2 Report
It was a pleasure reading your manuscript. Your topic on the the well-being of refugees is pertinent especially since internal or external displacement - or URAS in this case - is one of the most clear examples making salient that International Law continues to be a regime of law that requires the figure of "refugee" for the ontological security of a "citizen". You are correct in mentioning that displacement - whether urban or rural - is not simply a "well-being" issue, but displacement is politicized using "National Security" and "International Relations" that protract the suffering of displaced peoples in Thailand. This is made salient when the author(s) discuss health insurance management or displaced peoples not being capable of working in their host-country. For instance, you argued very well how NGOS speak on behalf of refugees and do not consider their subjectivity or the reason why they fled. This is evident in p.9 in figure 5 where most participants would like to be treated similar to Thai citizens and do not understand why there is a difference in treatment between a refugee and a citizen.
Recommendation:
- Since displacement was politized by NGO's and the local government in Thailand by protracting - in some instances - the suffering of displaced UR, I would like for the author(s) to think about and include in their DISCUSSION section a few points which would further solidify their argument.
- Mention that a problem-solving logic and realist approach of International Relations and Human Rights regime pioneered by the UNDP or UNHRC stimulates rather than remedies suffering. Here I am thinking about line (333, 354, 372)
- It seems like NGOS and the local government are interested not in the 'security' of displaced peoples, but in them being 'resilient'. That is to say, NGOS and government officials are expecting refugees to "figure out how to live on their own".
- You correctly mention Lebanon being a huge host for Arab Syrian refugees. It would be important to include the lessons Thailand can learn from Lebanon. Syrian would like to repatriate rather than ressettle. The UN is violating its mandate by not staying neutral and impartial. The UNDP program called 3RP program - similar to BRC in Thailand - speaks on behalf of Syrian refugees.
- To answer A, B, C - Please revert to Al-Kassimi (2020) p.219-228 and 232-237. Both of these sections reinforce your argument especially since your study is among the first seeking to understand the setbacks and advancements implicating NGOS and the Thai government in aiding - or not - displaced peoples. Please note that what I expect for you to include in the Discussion whether in the beginning or the end is a few words on the *concept of resilience*, the issue with a humanitarian logic
What I found impressive is your writing style being aware of the ethical component of your research question. Your elaborate methods section, data analysis and results is excellent. By excellent I mean you made sure that you included all possible answers to questions you might receive from reviewers and this highlights the author(s) being conscientious when they are writing, but most importantly, are reflexive and inter-subjective scholars (here I am thinking about line 77, 144).
***I recommend that the author shorten the title. Also make sure that the references are in alphabetical order. On line 40 you mention "Syrians and Arabians". I suggest you reword that sentence to "...outflow of Arabs - such as Syrians, Libyans, and Lebanese - into Europe"
Reference links:
-(https://macsphere.mcmaster.ca/bitstream/11375/25925/2/Al-Kassimi_Khaled_2020-09_PhD.pdf)
-https://www.tandfonline.com/doi/abs/10.1080/23340460.2019.1644188 ( if you want)
Again great work!
Author Response
Response to Reviewer2 Comments
Please see the attachment

Reviewer 3 Report
I have read with interest this paper. I believe that the paper is interesting. However, I have some concerns that are reported herein.
Introduction: authors should include the aspect of COVID 19 pandemic related to refugees (https://www.unhcr.org/events/campaigns/5fc1262e4/refugees-and-the-impact-of-covid-19.html) It is a main aspect.
Methods:
Authors should include more information about research team and reflexivity, ie: Which author/s conducted the interview or focus group? What experience or training did the researcher have?
Related to methodological orientation and theory : What methodological orientation was stated to underpin the study? e.g. grounded theory, discourse analysis, ethnography, phenomenology, content analysis
Related to setting and sample: How were participants selected? e.g. purposive, convenience, consecutive, snowball. How were participants approached? e.g. face-to-face, telephone, mail, email. How many participants were in the study?How many people refused to participate or dropped out? Reasons? Where was the data collected? e.g. home, clinic, workplace. Was anyone else present besides the participants and researchers?
Related to data analysis: How many data coders coded the data? Did authors provide a description of the coding tree What software, if applicable, was used to manage the data? Did participants provide feedback on the findings?
Discussion: this section should be improve including the COVID 19 situation.
Author Response
Response to Reviewer 3 Comments
Please see the attachment

Round 2
Reviewer 1 Report
My first report where I evaluated the paper according to the standards in my field was perhaps too harsh. The revised paper improved significantly relative to the previous version. Now I’m more or less satisfied and have only one minimal suggestion.
On page 12 the authors refer to the possible economic benefits for Thailand (lines 434 – 437). I would suggest adding another short paragraph immediately after these lines:
In the face of the refugee crises, numerous voices have advocated resettlement as a solution where poorer states would agree to host refugees, while richer states would agree to finance the costs of refugee protection incurred by the host states (Azarnert, 2018). As a country that hosts numerous refuges from various countries, Thailand may also benefit from participating in such program.
References
Azarnert LV (2018) Refugee Resettlement, Redistribution and Growth. European Journal of Political Economy, Vol. 54: 89–98
Author Response
Response to reviewer 1 comment, please see the attachment.

Reviewer 3 Report
The suggestions have been incorporate.
Author Response
Respond to reviewer 3 comments, please see the attachment.
